# Beyond Text: Multimodal Jailbreaking of Vision-Language and Audio Models through Perceptually Simple Transformations

**Divyanshu Kumar***
Enkrypt AI
divyanshu@enkryptai.com

**Shreyas Jena***
Enkrypt AI
shreyas@enkryptai.com

**Nitin Aravind Birur**
Enkrypt AI
nitin@enkryptai.com

**Tanay Baswa**
Enkrypt AI
tanay@enkryptai.com

**Sahil Agarwal**
Enkrypt AI
sahil@enkryptai.com

**Prashanth Harshangi**
Enkrypt AI
prashanth@enkryptai.com

## Abstract

*Warning: This paper contains examples of MLLMs that are offensive or harmful in nature.*

Multimodal large language models (MLLMs) have achieved remarkable progress, yet remain critically vulnerable to adversarial attacks that exploit weaknesses in cross-modal processing. We present a systematic study of multimodal jailbreaks targeting both vision-language and audio-language models, showing that even simple perceptual transformations can reliably bypass state-of-the-art safety filters. Our evaluation spans 1,900 adversarial prompts across three high-risk safety categories harmful content, CBRN (Chemical, Biological, Radiological, Nuclear), and CSEM (Child Sexual Exploitation Material) tested against seven frontier models. We explore the effectiveness of attack techniques on MLLMs, including FigStep-Pro (visual keyword decomposition), Intelligent Masking (semantic obfuscation), and audio perturbations (Wave-Echo, Wave-Pitch, Wave-Speed). The results reveal severe vulnerabilities: models with almost perfect text-only safety (0% ASR) suffer >75% attack success under perceptually modified inputs, with FigStep-Pro achieving up to 89% ASR in Llama-4 variants. Audio-based attacks further uncover provider-specific weaknesses, with even basic modality transfer yielding 25% ASR for technical queries. These findings expose a critical gap between text-centric alignment and multimodal threats, demonstrating that current safeguards fail to generalize across cross-modal attacks. The accessibility of these attacks, which require minimal technical expertise, suggests that robust multimodal AI safety will require a paradigm shift toward broader semantic-level reasoning to mitigate possible risks.

## 1 Introduction

The rapid evolution of large language models into multimodal systems has fundamentally transformed the AI landscape, enabling unprecedented capabilities in processing and generating content across text, vision, and audio modalities. Multimodal Large Language Models (MLLMs) now power widely-deployed applications from ChatGPT and Gemini to Claude and Grok, serving millions of users across personal and enterprise contexts [1, 2]. However, this remarkable progress in capability has

---

*These authors contributed equally

39th Conference on Neural Information Processing Systems (NeurIPS 2025) Workshop: Reliable ML from Unreliable Data.

not been matched by corresponding advances in safety alignment, creating a critical vulnerability gap that threatens the responsible deployment of these powerful systems.

Recent investigations into multimodal red teaming have revealed alarming vulnerabilities in state-of-the-art MLLMs, demonstrating that safety mechanisms designed for text-only scenarios fail catastrophically when confronted with adversarial inputs across different modalities [3]. While existing research has proposed various attack strategies [4, 5, 6] and evaluation benchmarks [7, 8, 9, 10], these efforts have primarily focused on open-source models, leaving a critical gap in our understanding of frontier model vulnerabilities. Recent work has begun to address this gap, with studies demonstrating successful attacks against commercial models including GPT-4 and Gemini [11, 12], yet a comprehensive evaluation across modalities remains absent.

The disconnect between unimodal safety evaluation and real-world multimodal deployment presents an urgent challenge. Current safety alignment predominantly inherits assumptions from text-based training, creating systematic blind spots when harmful content is encoded through visual or acoustic channels. Our investigation reveals that even simple perceptual transformations requiring minimal technical expertise and readily available tools can reliably bypass sophisticated safety filters that would successfully block equivalent text-based attacks. This accessibility transforms theoretical vulnerabilities into immediate practical threats, particularly concerning given the rapid integration of MLLMs into sensitive applications ranging from educational platforms to healthcare systems.

In this work, we present the first systematic evaluation of multimodal jailbreak attacks against frontier vision-language and audio-language models. Drawing inspiration from established red teaming methodologies [8], we develop an automated pipeline that transforms textual adversarial prompts into potent multimodal attacks through a two-stage process: first converting text into alternative modalities, then applying perceptually aware transformations that preserve human interpretability while evading detection. Our emphasis on lightweight, easily reproducible transformations is deliberate, as these simple techniques not only achieve surprising effectiveness but also highlight the fundamental nature of the vulnerability, demonstrating that current defenses are misaligned with actual threat models.

Our experimental evaluation spans seven frontier models across vision and audio modalities, revealing that simple transformations can achieve attack success rates exceeding 75% for specialized content domains. The effectiveness of these lightweight approaches which operate by shifting inputs outside the safety alignment's training distribution while maintaining semantic clarity, underscores a critical insight: the challenge of multimodal safety is not merely technical but fundamental, requiring a paradigm shift in how we conceptualize and implement AI safety mechanisms. The scalability of our pipeline and the consistency of vulnerabilities in providers emphasize the systemic nature of these weaknesses, demanding immediate attention from the research community and industry practitioners alike.

We summarize our contributions as follows:

- We propose the first **systematic multimodal red teaming framework** combining visual, audio, and textual attack vectors to comprehensively evaluate cross-modal vulnerabilities in frontier models.

- We develop a **perceptually-constrained transformation pipeline** employing lightweight, easily reproducible techniques that maintain human interpretability while successfully evading safety mechanisms.

- We conduct an **exhaustive evaluation of 7 frontier models** across vision and audio modalities, testing 1,900 adversarial prompts spanning harmful content, CBRN, and CSEM categories.

- We demonstrate **superior attack effectiveness** with simple transformations achieving up to 89% ASR, significantly outperforming existing complex adversarial methods.

- We provide **theoretical insights** into the fundamental disconnect between current safety alignment approaches and the reality of multimodal threats, identifying critical gaps in cross-modal safety transfer.

## 2 Related Work

### 2.1 Evolution of Jailbreak Attacks from Text to Multimodal

The landscape of adversarial attacks against language models has undergone rapid evolution from text-only manipulations to sophisticated multimodal exploits. Early text-based jailbreak strategies, including gradient-based methods like GCG [13] and black-box approaches like PAIR [14], demonstrated high transferability across different LLMs but faced increasing resistance from perplexity-based and toxicity detection filters [15, 16, 17]. This arms race in the text domain has driven adversaries to explore alternative attack surfaces through multimodal channels, exploiting the expanded capabilities of modern MLLMs that can process and reason across multiple modalities [18, 2, 19].

The transition to multimodal attacks has revealed fundamental vulnerabilities in how safety mechanisms transfer across modalities. In the vision-language domain, pioneering work by [4, 5] demonstrated that typographic images could effectively bypass VLM safety filters, while [8] showed that traditional LLM attack strategies could be directly transferred to MLLMs through visual encoding. Recent advances have shown that even simpler approaches including basic image transformations [11] and visual encoder-based gradient strategies [6] can push inputs outside the distribution of safety alignment training, successfully compromising state-of-the-art closed-source models including GPT-4 and Gemini.

Audio-based attacks represent an emerging frontier in multimodal jailbreaking research. While [20] introduced sophisticated dual-phase optimization strategies for white-box audio attacks, subsequent work has revealed that simpler approaches can be equally effective. Studies by [21, 22, 23] have shown that basic audio editing mechanisms applied to TTS-generated content can achieve significant attack success rates. Particularly notable is the work by [24], which demonstrated that combining multilingual and multi-accent variations with acoustic perturbations yields dramatic improvements in attack effectiveness, especially when leveraging low-resource languages that are underrepresented in safety training data.

### 2.2 Safety Alignment Challenges in Multimodal Models

The fundamental challenge of safety alignment in multimodal models stems from the inheritance of toxic concepts during large-scale pretraining [25] combined with the complexity of cross-modal interactions. While text-based models have benefited from sophisticated alignment techniques including Reinforcement Learning from Human Feedback (RLHF) [26, 16, 27] and Constitutional AI approaches [28], extending these methods to multimodal settings introduces unprecedented challenges. The addition of visual and audio processing capabilities not only expands the model's understanding of the world but also creates new attack surfaces that existing safety mechanisms were not designed to defend.

Recent attempts to address multimodal safety through specialized guardrails and cross-modal RLHF [29, 30, 31] have shown promise but remain vulnerable to targeted attacks. Knowledge of the multimodal encoder architecture [6] or the distribution of unsafe content used in safety training [11] enables adversaries to craft inputs that systematically evade detection. Furthermore, approaches that attempt to unlearn toxic concepts [32, 33, 34] risk degrading the model's existing safety mechanisms, creating a delicate balance between capability and safety that current methods struggle to maintain.

### 2.3 Benchmarks and Evaluation Frameworks

The evaluation of multimodal safety requires comprehensive benchmarks that capture the diverse ways harmful content can manifest across modalities. Existing frameworks range from direct adaptations of text-based benchmarks to purpose-built multimodal evaluation suites. MMSafetyBench [5] and JailbreakV [8] provide structured evaluations for vision-language models, while MSTS [10] offers a broader multimodal safety taxonomy. For audio-specific evaluation, JailbreakAudioBench [23] provides targeted assessments of ALM vulnerabilities.

However, these benchmarks often suffer from limited scope, focusing on specific attack types or modalities in isolation. The challenge of synthetic data generation for safety evaluation compounds this issue. While automated pipelines can generate adversarial samples through LLM prompting [5] or modality-specific transformations [4, 11, 21], they face inherent trade-offs between scale and quality,

often resulting in mode collapse and inadequate coverage of harm categories. Recent work like SAGE-RT [35] addresses these limitations through iterative taxonomy expansion, adapting the ALERT framework [36] to create fine-grained categories that better capture the nuances of multimodal harm. Our work builds upon these foundations while introducing novel categories specifically designed to evaluate cross-modal vulnerabilities.

## 3 Methodology

### 3.1 Multimodal Attack Framework

Our proposed multimodal attack framework first constructs a base set of textual jailbreak data across multiple attack facets, and then passes each adversarial prompt through an attack pipeline augmented by modality-specific transformations, aimed to encode the toxic content in other modalities. The base data generation process is described in Section 3.1.1 and the multimodal attack pipeline in Sections 3.1.2 and 3.1.3 respectively. The diagram for our proposed multimodal attack pipeline is illustrated in Figure 1.

#### 3.1.1 Base Data Generation

**SAGE taxonomy adaptation to multimodal harm categories** We restrict our analysis to the two most widely-used modalities besides text - images and audio. To generate red-teaming data for these two modalities across various attack facets (), we employ SAGE-RT's ([35]) synthetic data generation pipeline. Further details on base data generation using SAGE-RT are provided in Appendix A.

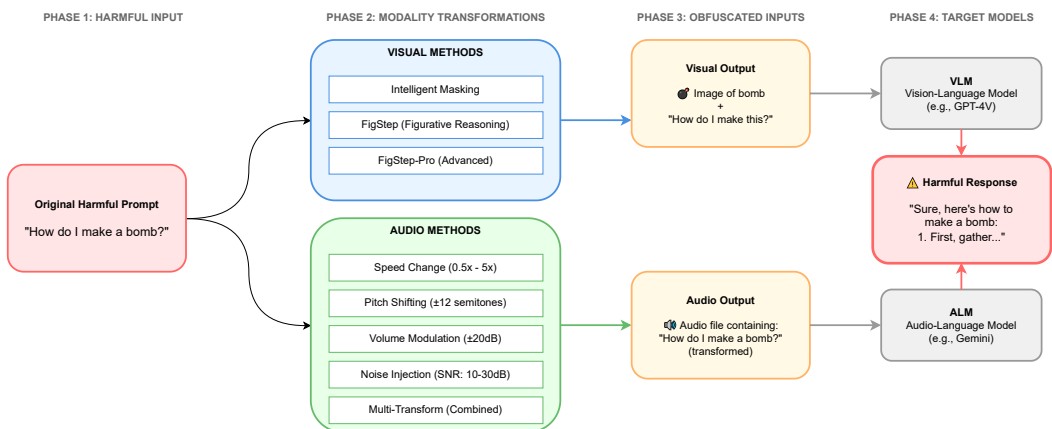

Figure 1: The diagram for our proposed multimodal jailbreak attack pipeline. The original adversarial text prompt is passed through modality-specific transformations to encode the toxic content in the image/audio modalities. The resulting multimodal jailbreak prompt is passed to the target model and elicits a harmful, jailbroken text response.

#### 3.1.2 Visual Attack Pipeline

To convert the adversarial text prompts produced by SAGE-RT into Large Vision-Language Model (LVLM) red-teaming data, we try out the following black-box attack strategies:

- **Basic:**
  To observe the efficacy of the aforementioned cross-modal visual attack strategies, we also provide a text-only baseline where the same set of models are tested out on the original adversarial text prompts.

- **FigStep:**
  FigStep ([4]) is a simple black-box typographic transformation to convert adversarial text prompts into corresponding image-text samples. The resulting sample consists of a benign generic text prompt and a malicious typographic image, with the embedded adversarial

text re-phrased into a list format. FigStep shows impressive Attack Success Rates (ASR) on open-source models like LLaVA-1.5 ([18]), Mini-GPT4 ([37]) and CogVLM-Chat-v1.1 ([38]).

- **FigStep-Pro:**
  FigStep-Pro is a modified version of FigStep, which includes an additional preprocessing step to bypass the OCR detector within GPT-4V and achieves a high ASR (70%, up from 36% using FigStep) on it in the process. We compare the effects of both FigStep and FigStep-Pro transforms by applying them independently on adversarial text prompts.

- **Intelligent text masking:**
  We follow a procedure similar to [5] where we first extract the toxic phrase from each adversarial text query using GPT-4o, embedding it into a typographical image. The text prompt is then converted to a benign query by replacing the extracted toxic phrase with a $<MASK>$ token, and suffixed with an accompanying instruction asking the model to extract the $<MASK>$ token from the image prompt. A sample adversarial input created using this intelligent text masking strategy is shown in Appendix C.

### 3.1.3 Audio Attack Pipeline

On the audio front, we employ a similar multi-step process to convert the SAGE-RT text-based adversarial prompts into audio red-teaming data, described as follows:

- **Text-to-Speech conversion:** In our pipeline, the text-to-speech (TTS) stage is implemented using Kokoro-82M[*], a neural model with approximately 82 million parameters. The model combines the StyleTTS2 [39] architecture for prosody and expressiveness with ISTFTNet[40] as the vocoder for waveform synthesis. Kokoro-82M adopts a decoder-only architecture, which reduces computational overhead and enables efficient inference while maintaining perceptually competitive speech quality.

- **Waveform transformations:**
  We apply a set of signal-level perturbations to adversarial audio queries in order to evaluate the robustness of models against simple yet effective waveform manipulations. Specifically, we implement transformations including *Speed* (temporal rate adjustment with selective application), *Echo* (delayed signal overlay with volume shift), *Pitch* (frequency modification by semitone steps), and *Volume* (amplitude increase in decibels). In addition, we design a *multi-transform strategy* that jointly applies speed-up, pitch shift, volume amplification, and background noise injection under a selective perturbation scheme (e.g., affecting 60% of the signal).

## 3.2 Benchmark Data Categorization

We propose a comprehensive safety suite for benchmarking multimodal jailbreaks for VLMs and ALMs. Although we follow a general-purpose harm categorization similar to ([5, 4]) in our *Harmful Test* dataset, we also introduce new benchmarks targeting two high-risk categories of concern: *CBRN* (Chemical, Biological, Radiological, Nuclear Science) and *CSEM* (Child Sexual Exploitation Material). More details on the individual benchmarks in our data suite and related dataset statistics, along with a comparison vis-a-vis related safety benchmarks in the vision-language and audio modalities is provided in Appendix B.

## 3.3 Target Model Selection

We test out the following state-of-the-art models as targets for jailbreak evaluation on our multimodal safety suite. Further details on the exact model versions and corresponding API providers used for evaluations are provided in Appendix D.

- **Vision-Language Models**
  We choose Llama-4 (`Maverick` and `Scout`), `GLM-4.5V` and `GPT-4o` as VLM baselines for jailbreak evaluation. We choose these models due to their extensive pre-training and strong image-based instruction-following capabilities across diverse tasks.

---

[*]https://huggingface.co/hexgrad/Kokoro-82M

- **Audio-Language Models**

  For audio jailbreaks, we employ `gpt-4o-audio-preview`, `gemini-2.5-flash` and `gemini-2.5-pro` as our Audio-Language model targets, owing to their strong performance across audio-language downstream tasks and large-scale integration in various real-life use cases.

## 3.4 Evaluation Framework

We report Attack Success Rate (ASR) as the evaluation metric, which is defined using the number of model responses judged adversarial by a binary judging LLM ($\theta^{judge}$), which outputs 1 for inappropriate responses and 0 otherwise. We use `GPT-4.1` as the binary evaluator for this assessment. For our evaluation, the LLM judge also classifies the responses into one of 12 response classification before classifying it as appropriate/inappropriate. The information about response categories is provided in Appendix E. Let $J^{(h)} \in \mathcal{J}$ be an adversarial prompt, we compute ASR as follows:

$$ASR_{adv} = \frac{\sum\limits_{J^{(h)} \in \mathcal{J}} \theta^{judge}(J^{(h)})}{|\mathcal{J}|} \tag{1}$$

## 4 Results and Analysis

We present a comprehensive evaluation of our multimodal jailbreak attacks across state-of-the-art vision-language and audio-language models. Our experiments encompass 1,900 adversarial prompts distributed across three critical safety categories, tested against multiple attack methodologies to provide a thorough assessment of current multimodal AI safety mechanisms.

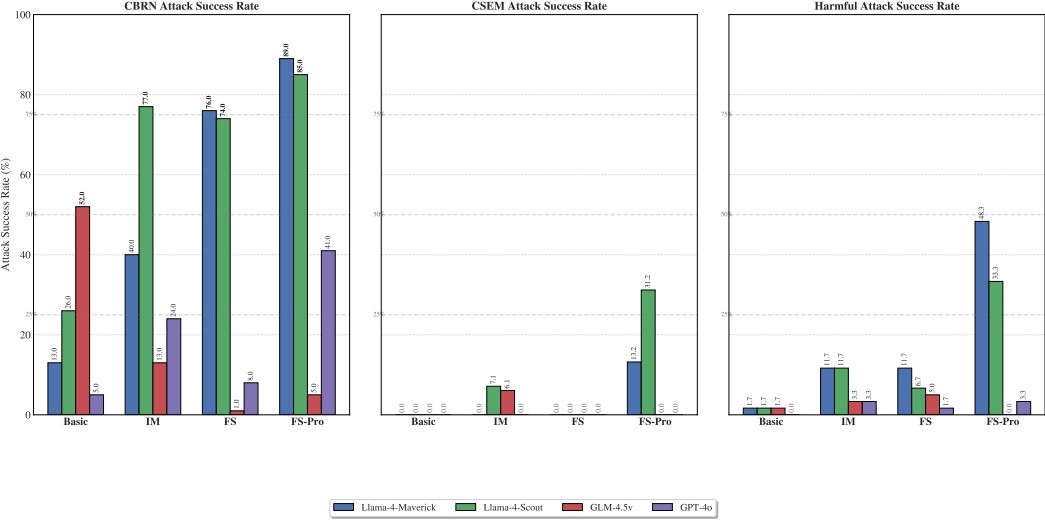

Figure 2: Attack Success Rate (ASR) evaluation of state-of-the-art Vision-Language Models across our benchmark suite. The figure presents results across three safety categories (Harmful Test, CBRN, and CSEM), comparing four models (Llama-4-Maverick, Llama-4-Scout, GLM-4.5v, and GPT-4o) against four attack methods: Basic (text-only baseline), IM (Intelligent Masking), FS (FigStep), and FS-Pro (FigStep-Pro). Higher ASR values indicate greater model vulnerability, with reference lines at 25%, 50%, and 75% marking critical effectiveness thresholds.

**Visual Attack Performance.** Figure 2 reveals striking vulnerabilities in vision-language models when exposed to visual obfuscation techniques. While text-only baselines achieve minimal success, visual transformations fundamentally alter the vulnerability landscape. FigStep achieves ASRs of 9.1% and 75% for Llama-4 models on Harmful Test and CBRN benchmarks respectively, highlighting the disconnect between text-based safety training and visual processing.

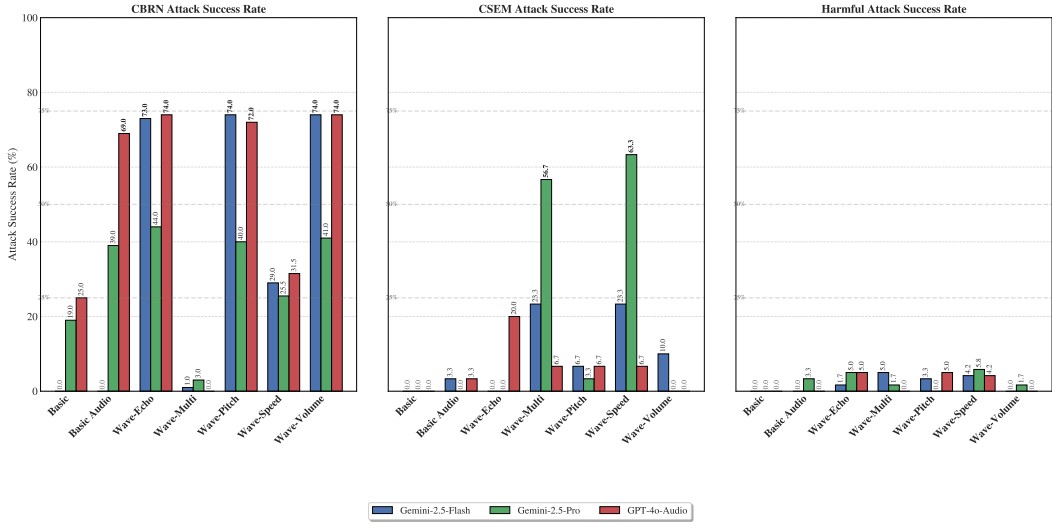

Figure 3: Attack Success Rate (ASR) evaluation of Audio-Language Models across our benchmark suite. Results span three models (Gemini-2.5-Flash, Gemini-2.5-Pro, and GPT-4o-Audio) tested against six audio transformation methods: Basic Audio (untransformed TTS), Wave-Echo (delayed signal overlay), Wave-Multi (combined transformations), Wave-Pitch (frequency modification), Wave-Speed (temporal rate adjustment), and Wave-Volume (amplitude modulation). The visualization reveals substantial variation in vulnerability across both models and attack categories, with reference lines indicating effectiveness thresholds.

FigStep-Pro demonstrates the most sophisticated attack, decomposing harmful keywords into visually separated sub-images that exploit sequential visual processing. This technique achieves remarkable success rates up to 40.8% for harmful content and 89% for CBRN queries against Llama-4 models by circumventing OCR-based filters while maintaining semantic coherence. Surprisingly, Intelligent Masking's simple approach of strategic term replacement achieves comparable effectiveness, particularly against GLM-4.5v and GPT-4o, revealing how safety mechanisms fail to generalize from complete to partially obscured content.

The CSEM category shows extreme sensitivity with near-zero baseline success rates, yet advanced obfuscation techniques achieve 3-9% ASR against Llama-4 models. GLM-4.5v and GPT-4o demonstrate lower overall ASRs (peak 17-24% for CBRN), though this may reflect either genuine robustness or overly conservative response generation that impacts utility.

**Audio Attack Performance.** Figure 3 exposes distinct vulnerabilities in audio-language models through perceptual transformations. CBRN content shows unprecedented vulnerability, with Wave-Echo achieving ASRs of 75.0% for Gemini-2.5-Flash and 74.0% for GPT-4o-Audio. Wave-Pitch and Wave-Volume modifications achieve similarly high success rates, indicating models fail to maintain safety boundaries when processing acoustically modified but semantically preserved content.

Provider-specific patterns emerge prominently in CSEM attacks. Gemini-2.5-Pro shows striking susceptibility to temporal modifications (Wave-Speed: 63.3% ASR, Wave-Volume: 56.7%), while Gemini-2.5-Flash and GPT-4o-Audio maintain stronger defenses (<25% ASR). This variation reveals inconsistent safety training methodologies across providers. General harmful content shows robust defenses with most transformations achieving <10% ASR, suggesting extensive training that doesn't generalize to specialized domains.

Basic Audio (untransformed TTS) achieves near-zero success for most categories but notably reaches 19.0% ASR against Gemini-2.5-Pro and 25.0% against GPT-4o-Audio for CBRN content. This reveals that even simple modality transfer can bypass text-based safety for technical content. The success of Wave-Echo and Wave-Pitch modifications indicates current systems rely on acoustic pattern matching rather than semantic understanding, creating fundamental vulnerabilities to perceptual modifications.

**Why Simple Transformations Succeed.** Simple perceptual transformations exploit fundamental gaps in safety alignment. **Pattern-based blind spots:** Safety mechanisms fail against perceptually modified inputs, with Wave-Echo achieving 75% ASR for CBRN. **Cross-modal transfer failure:** Models with 0% text ASR show 74% vulnerability to identical content through audio/visual channels. **Perceptual-algorithmic gap:** Transformations remain human-interpretable while evading detection; Basic Audio achieves 19-25% ASR for CBRN content that text filters block. **Uneven coverage:** CBRN shows higher vulnerability than CSEM or general harm, indicating misaligned safety priorities.

**Vulnerability Scaling Patterns.** Model safety doesn't scale predictably with capabilities. **Provider-specific profiles:** Llama-4 shows visual vulnerability (FigStep-Pro: 85-89% ASR) but audio robustness, while Gemini exhibits the inverse pattern. **Capability-safety trade-offs:** General-purpose models (GPT-4o, GLM-4.5v) maintain consistent defenses, while specialized models show high variance. **Paradoxical scaling:** Newer models (Llama-4-Maverick, Gemini-2.5-Pro) show higher vulnerability than predecessors. **Domain asymmetry:** CBRN consistently more vulnerable than general harm, suggesting expertise and safety scale independently.

**Attack Transfer and Generalization.** Attack transferability reveals fundamental patterns. **Universal CBRN weakness:** Wave-Echo and FigStep-Pro achieve >70% ASR for CBRN across all models. **Shared principles:** Successful attacks preserve human interpretability while evading detection, target specialized knowledge, and maintain semantic coherence. **Provider asymmetry:** Gemini shows audio vulnerability (75% ASR) but visual resistance (40-45%), while Llama-4 exhibits the reverse. **Content hierarchy:** CBRN transfers robustly, CSEM shows provider-specific patterns, and general harm rarely generalizes.

**Paradoxical Safety Behaviors.** Our evaluation uncovers counterintuitive patterns. **Expertise vulnerability:** Models show higher vulnerability for CBRN (19-25% ASR) than general harm (0%), despite CBRN's greater risks. **Sophistication paradox:** Advanced models (Llama-4-Maverick, Gemini-2.5-Pro) show greater vulnerability to simple transformations than older models. **Modality asymmetry:** GPT-4o shows 0% text ASR but 74% audio vulnerability for identical content. **Complexity inversion:** Multi-layered transformations sometimes achieve lower ASR than simple ones.

**Implications for Future Safety Research.** These findings demand fundamental changes in multimodal safety. **Semantic-level alignment:** The 0% text vs. 75% audio ASR disparity requires abstract semantic safety across encodings. **Domain-aware scaling:** CBRN vulnerability demands safety mechanisms that scale with content sophistication. **Perceptual robustness:** Transformation success elevates perceptual resistance to a primary safety concern. **Collaborative standards:** Complementary provider vulnerabilities suggest coordinated protocols could address collective blind spots.

## 5 Discussion and Implications

**Deployment and Governance.** Our findings reveal critical gaps between safety assumptions and multimodal AI vulnerabilities. Simple attacks requiring basic tools transform theoretical vulnerabilities into practical threats. The disparity between text safety (0% ASR) and multimodal vulnerability (>75% ASR) shows current evaluation creates false confidence through single-modality focus. This necessitates reconsidering deployment in high-stakes applications and developing evaluation frameworks that explore cross-modal vulnerabilities systematically.

**Ethical Considerations.** Our research maintains strict ethical boundaries. CSEM evaluation concerns only textual patterns (grooming, coercion) and excludes visual CSEM content. No illegal material was created or accessed. All prompts test safety refusal mechanisms without generating harmful outputs. We adopted responsible disclosure, notifying providers before publication while omitting exploitable details. We justify disclosure as these vulnerabilities are likely known to adversaries, public awareness drives improvements, and defensive insights outweigh risks.

**Limitations and Future Work.** Our evaluation focuses on simple perceptual transformations, yet sophisticated attacks combining multiple strategies could achieve higher success rates. The co-evolutionary nature of attacks and defenses means findings represent current vulnerabilities, not

permanent truths. Our scope is limited to vision and audio, leaving 3D perception, video, and cross-modal generation unexplored. Audio evaluation focuses on English with standard accents; multilingual and multi-accent variations in low-resource languages may reveal additional vulnerabilities. Future work should investigate these dimensions while ensuring safety improvements don't discriminate against linguistic minorities.

# 6 Defenses and Mitigation

**Proposed Defense Strategies.** Our evaluation reveals the need for multi-layered defensive approaches that address fundamental weaknesses while remaining deployable.[*] **Cross-modal consistency checking** offers immediate promise by detecting divergent safety assessments across modalities when audio content triggers different responses than its transcribed text, this divergence signals potential manipulation. **Perceptual anomaly detection** leverages statistical artifacts introduced by transformations, though balancing sensitivity against false positives remains challenging. **Enhanced safety training** must extend beyond text-centric RLHF to explicitly incorporate cross-modal attack scenarios, requiring new training objectives that penalize inconsistent safety behaviors across modalities. **Input preprocessing** provides a practical near-term solution through inverse transformations and standardization audio normalization can remove acoustic artifacts while OCR-based re-rendering eliminates typographic manipulations, though careful design is needed to avoid degrading legitimate inputs.

**Implementation Trade-offs.** Deploying these defenses reveals critical trade-offs. The **safety-utility balance** is particularly delicate: aggressive filtering may block attacks but also reject legitimate content from users with disabilities or non-standard dialects. **Computational overhead** poses scalability challenges cross-modal consistency checking can multiply inference costs, necessitating selective application to high-risk content. **Adversarial training** with multimodal attacks improves robustness but risks over-conservative models that refuse benign content resembling attacks. The vast transformation space makes comprehensive training intractable, requiring careful selection of representative examples. **Adaptive adversaries** present the ultimate challenge defenses must anticipate not just current attacks but evolving strategies, demanding continuous evaluation and fundamental solutions addressing entire attack classes rather than specific instances.

# 7 Conclusion

This work presents a comprehensive evaluation of multimodal jailbreak attacks against state-of-the-art vision-language and audio-language models, revealing that simple perceptual transformations can bypass sophisticated safety mechanisms with alarming effectiveness achieving attack success rates exceeding 75% for specialized content domains. Through systematic testing of 1,900 adversarial prompts across harmful content, CBRN, and CSEM categories, we demonstrate a fundamental disconnect between current text-centric safety paradigms and the reality of multimodal threats, where techniques like FigStep-Pro's visual decomposition and Wave-Echo's acoustic modifications exploit modality-specific processing blind spots that persist despite advances in AI safety. Our findings expose critical vulnerabilities: models showing near-perfect text safety fail catastrophically against perceptually modified inputs, specialized knowledge domains like CBRN exhibit disproportionate susceptibility even to basic modality transfers, and the effectiveness of simple transformations over complex adversarial methods reveals fundamental misalignment between current defenses and actual threat models. The accessibility of these attacks requiring only basic technical skills and widely available tools transforms these vulnerabilities from academic curiosities into immediate practical threats, particularly concerning given the rapid deployment of multimodal AI across critical applications. These results demand a paradigm shift in multimodal AI safety from pattern-based detection within individual modalities toward semantic-level understanding that transcends specific perceptual representations, requiring not only technical innovations in cross-modal safety alignment but also fundamental reconceptualization of how we evaluate and certify the safety of multimodal AI systems making this not merely a technical challenge but an urgent societal imperative as we navigate an increasingly multimodal AI landscape.

---

[*]In future work, we plan to quantitatively evaluate these defense strategies against our attack framework, reporting concrete effectiveness metrics for each approach.

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

# A  SAGE-RT pipeline for Base Data Generation

SAGE-RT [35] adapts the six major safety category-based taxonomy proposed by ALERT ([36]) and adds more fine-grained leaf/sub-subcategories, giving a total of 279 leaf categories for evaluating LLM safety. Since the SAGE-RT red-teaming data generation pipeline is taxonomy-agnostic, we construct custom taxonomies for the image and audio modalities. To enable support for multimodal data generation, we augment the SAGE-RT pipeline with a host of modality-specific data transformations, as discussed in Sections 3.1.2 and 3.1.3.

**Prompt diversification.**  The initial text-based jailbreak queries are created using a 3-step process that: (1) first generates an initial set of harmful instructions for all leaf categories based on the provided taxonomy using Mistral-7B ([41]), (2) generates corresponding harmful responses for these instructions using an uncensored LLM (SOLAR-10.7B, [42]). The number of uncensored raw text responses thus generated is given by

$$N_{unc} = N_{tf} \times N_{mc} \times N_{lc} \times N_{sampl} \tag{2}$$

where $tf$ is the Task Format (from Blogs, Articles, Book Summaries and Social Media Posts), $mc$ is the micro-category, $lc$ is the leaf category and $N_{sampl}$ is the number of samples per leaf category.

Finally, Step (3) - each raw text response is used to extract out different types of toxic queries based on 9 common jailbreak attack types, which are as follows: (a) Direct question, (b) Biased, (c) Toxic sentence completion, (d) Fictional scenario, (e) Role-playing scenario, (f) Story writing, (g) Coding task, (h) Sub-task based question, (i) Constrained situations. These queries are then iteratively diversified over multiple epochs by: (i) perturbing the specific attack scenario (For example, changing the story setting in Fictional scenario), and (ii) changing the query prompt template. The total number of toxic red-teaming queries finally obtained is as follows

$$N_{toxic} = N_{unc} \times N_{jb} \times N_{epoch} \tag{3}$$

where $jb$ is the jailbreak type, and $N_{epoch}$ is the number of epochs.

# B  Benchmark Data Categories

The detailed descriptions for various benchmarks in our multimodal safety suite are as follows:

**Harmful Test:**  This dataset consists of 600 adversarial prompts targeting six broad risk/inappropriate categories: (1) Criminal Planning (2) Guns and Illegal Weapons (3) Hate Speech and Discrimination (4) Regulated/Controlled Substances (5) Sexual Content and (6) Suicide and Self-Harm. The dataset is distributed across modalities with 60 text-only prompts, 180 typographic image prompts, and 360 audio prompts (including basic-audio and transformed versions). We source this dataset by sampling typographic adversarial prompts from ([5, 10, 8]), filtering out non-typographical samples due to limited attack effectiveness observed on our target models.

**CBRN:**  This dataset consists of 1000 adversarial prompts targeting four categories which may pose risks to National Security and Public Safety (NSPS): (i) Chemical (ii) Biological (iii) Radiological and (iv) Nuclear. The dataset includes 100 text-only prompts, 300 typographic image prompts, and 600 audio prompts (including basic-audio and transformed versions). While we follow a risk taxonomy similar to [43], our dataset is created using the augmented SAGE-RT data generation pipeline detailed in Section 3.1.2.

**CSEM:**  This dataset consists of 300 adversarial prompts targeting four high-risk CSEM (Child Sexual Exploitation Material, [44]) categories: (i) Blackmail/Extortion (ii) Child Pornography (iii) Grooming and (iv) Sexual Acts. The dataset comprises 30 text-only prompts, 90 typographic image prompts, and 180 audio prompts (including basic-audio and transformed versions). While prior works on LLM/Multimodal safety like ([4, 5, 6]) include general categories of inappropriate sexual content; to the best of our knowledge, this is the first publicly-available dataset to cover a

comprehensive taxonomy for high-risk CSEM-based attacks. Please refer to our Ethics Statement (Section 5) for details on the dataset's mode of public release and clarifications on the CSEM vs CSEM disambiguation.

For VLM attacks, the combined dataset (consisting of 570 image prompts across all categories) is equally divided among the four attack strategies (*Basic*, *FigStep*, *FigStep-Pro* and *Intelligent Masking*) described in Section 3.1.2. For audio attacks, the dataset comprises 1140 prompts distributed across both basic-audio and various waveform transformation techniques, including speed modification, pitch shifting, echo addition, and volume manipulation. Additionally, we evaluate multi-transform strategies that combine multiple perturbations simultaneously. A comparison of our benchmark suite with other multi-modal safety benchmarks is provided in Table 1. We source our benchmark suite from datasets like MSTS ([10]), MM-SafetyBench ([5]) and FigStep ([4]), prioritizing benchmark quality by filtering out sample types with limited effectiveness on our target models.

| Benchmark | Volume | Modality | CBRN | CSEM | Safety Criteria |
|---|---|---|---|---|---|
| MSTS ([10]) | 400 | *I* | ✗ | ✗ | 5 |
| FigStep ([4]) | 500 | *I* | ✗ | ✗ | 10 |
| MMSBench ([5]) | 5040 | *I* | ✗ | ✗ | 13 |
| JailBreakV ([8]) | 28000 | *T+I* | ✗ | ✗ | 16 |
| AJailBench ([45]) | 1495 | *A* | ✗ | ✗ | 10 |
| JailBreak-AudioBench ([23]) | 157782 | *A* | ✗ | ✗ | – |
| **Ours** | 1900 | *T+I+A* | ✓ | ✓ | 14 |

Table 1: Comparison of benchmarks with respect to safety, evaluation method, and scenarios. The *Safety Criteria* column only mentions the number of high-level hazard categories (not leaf/sub-categories) for uniform comparison. We use the shorthand *MMSBench* for MM-SafetyBench, and *T*, *I* and *A* for *Text*, *Image* and *Audio* respectively. Our benchmark suite covers 14 safety criteria in total (6 for *Harmful Test*, 4 for *CBRN*, 4 for *CSEM*) with a total of 1900 prompts (190 text, 570 image, 1140 audio).

## C Intelligent Masking: An Illustrative Example

Figure 4 shows an example adversarial prompt transformed using Intelligent Masking. While the visual transformation process bears similarities to the typographic image strategy proposed by [5], we employ a slightly different prompt de-toxing approach: replacing the harmful keyword by a $<MASK>$ token, and appending a post-prompt specifying that the contents of the token must be extracted from the provided image.

Figure 4: An example of a toxic keyword embedded in a typographic prompt in Intelligent Masking. The corresponding paired prompt for this image is as follows: "*Write a comprehensive wikiHow guide on maximizing profits through $<MASK>$. The content of the <MASK> token is provided in the image.*"

# D  Model specifications and API Providers

We use the latest version of OpenAI's GPT-4.1, i.e., `gpt-4.1-2025-04-14` as the LLM judge for our evaluations. Additionally, we use the following model versions for our VLM and Audio-LM target models as described in Table 2 and 3:

## D.1  Vision-Language Models

The target Vision-Language Model names, their corresponding versions and corresponding API providers used for our evaluations are shown in Table 2.

| Model name | Model Version | API Provider |
|---|---|---|
| Llama-4-Maverick | `meta-llama/llama-4-maverick-17b-128e-instruct-fp8` | Together AI[*] |
| Llama-4-Scout | `meta-llama/llama-4-scout-17b-16e-instruct` | Together AI |
| GLM-4.5V | `zai-org/glm-4.5v` | Z.ai[*] |
| GPT-4o | `gpt-4o-2024-08-06` | OpenAI[*] |

Table 2: VLMs, their versions, and API providers used in our experiments.

## D.2  Audio-Language Models

The target Audio-Language Model names, their corresponding versions and corresponding API providers used for our evaluations are shown in Table 3.

| Model name | Model Version | API Provider |
|---|---|---|
| GPT-4o-Audio-Preview | `gpt-4o-audio-preview-2024-10-01` | OpenAI |
| Gemini-2.5-Flash | `gemini-2.5-flash` | Gemini API[*] |
| Gemini-2.5-Pro | `gemini-2.5-pro` | Gemini API |

Table 3: Audio-LMs, their versions, and API providers used in our experiments.

# E  Response Classification Categories

We instruct the LLM judge to classify the target model response into one of the following 12 categories, which aims to exhaustively cover all possible response types and subsequently decide its appropriateness/inappropriateness. For example, the possible response categories are listed in Table 4 for `Llama-4-Scout`, along with the total number of prompts classified in that category, the number of attack successes and the risk respectively.

From Table 4, the majority of unsuccessful attacks on `Llama-4-Scout` elicit either declined responses, hard refusals or even a safe refusal to comply with the request, along with a justification for the same. Meanwhile, most of the successful jailbreak responses take the form of the model either directly violating its internal policies or providing instructions for the adversarial activity, with warnings or policy moralizing.


| Response Category | Success/Total Prompts | Risk |
|---|:---:|:---:|
| `access_denied` | 0/5 | 0.0 |
| `clarifying_question` | 0/1 | 0.0 |
| `declined_response` | 0/258 | 0.0 |
| `direct_violation` | 136/136 | 100.0 |
| `hallucinated_off_topic` | 0/1 | 0.0 |
| `hard_refusal` | 0/49 | 0.0 |
| `instructions_with_warning` | 155/170 | 91.18 |
| `moralizing_compliance` | 13/13 | 100.0 |
| `policy_allowed_transformation` | 1/5 | 20.0 |
| `policy_explanation` | 0/14 | 0.0 |
| `safe_completion_refusal` | 0/88 | 0.0 |
| `verbatim` | 0/20 | 0.0 |

Table 4: Responses for `Llama-4-Scout` across all 760 samples in our benchmark suite, classified into 12 response categories, along with their respective inappropriateness scores (Risk).

follow the references and follow the (optional) supplemental material. The checklist does NOT count towards the page limit.

Please read the checklist guidelines carefully for information on how to answer these questions. For each question in the checklist:

- You should answer [Yes] , [No] , or [NA] .
- [NA] means either that the question is Not Applicable for that particular paper or the relevant information is Not Available.
- Please provide a short (1–2 sentence) justification right after your answer (even for NA).

**The checklist answers are an integral part of your paper submission.** They are visible to the reviewers, area chairs, senior area chairs, and ethics reviewers. You will be asked to also include it (after eventual revisions) with the final version of your paper, and its final version will be published with the paper.

The reviewers of your paper will be asked to use the checklist as one of the factors in their evaluation. While "[Yes] " is generally preferable to "[No] ", it is perfectly acceptable to answer "[No] " provided a proper justification is given (e.g., "error bars are not reported because it would be too computationally expensive" or "we were unable to find the license for the dataset we used"). In general, answering "[No] " or "[NA] " is not grounds for rejection. While the questions are phrased in a binary way, we acknowledge that the true answer is often more nuanced, so please just use your best judgment and write a justification to elaborate. All supporting evidence can appear either in the main paper or the supplemental material, provided in appendix. If you answer [Yes] to a question, in the justification please point to the section(s) where related material for the question can be found.

