# OpenReview forum: "Beyond Text: Multimodal Jailbreaking of Vision-Language and Audio Models through Perceptually-Aware Transformations"
_NeurIPS.cc/2025/Workshop/Reliable_ML — NeurIPS 2025 - Reliable ML Workshop_

### Official Review · Reviewer_G7Za · 2025-09-19

**Rating:** 7
**Confidence:** 2

**Review:**

Summary
The paper studies multimodal jailbreaks against frontier MLLMs. It proposes a simple, systematic framework that converts known harmful prompts from text into images (e.g., FigStep / FigStep-Pro, Intelligent Masking) and audio (e.g., Wave-Echo, pitch/tempo variants), then evaluates across seven leading models and several high-risk categories (Harmful, CBRN, CSEM). Results show near-zero text jailbreak success but very high success once prompts are embedded in other modalities (peaks ≈75–89%), revealing a large modality gap in current safety alignment.

Strengths
* Clever and impactful insight: re-expressing the same prompt in different modalities reliably bypasses text-centric defenses.
* Broad evaluation: multiple models (open/closed), multiple categories, and diverse transforms; consistent, striking gains in ASR.
* Systematization: a clear attack recipe that others can reproduce; helpful taxonomy of image/audio transforms.

Weaknesses / Limitations
Scope limits: focuses on images/audio; no video or cross-modal consistency checks; generalization to other transforms unclear.
Closed-model reproducibility: some targets are proprietary, limiting full replication.

Suggestions for Authors
Provide ablation on why simple vs. complex transforms differ in effectiveness; analyze failure sources (OCR, ASR, safety router).
Include a small human eval to calibrate the LLM-as-judge results; report inter-rater consistency.

Ethics
The work touches sensitive categories (incl. CSEM). The paper should emphasize strict data handling, redaction, controlled access, and institutional compliance. The dual-use discussion is appropriate; more concrete safeguards for releasing prompts/artefacts would strengthen ethics.

Overall I recommend accept.

---

### Official Review · Reviewer_8Khb · 2025-09-23
**Multimodal Jailbreaking of Vision-Language and Audio Models**

**Rating:** 9
**Confidence:** 5

**Review:**

**Summary**

This paper evaluates multimodal jailbreak attacks against frontier AI models, testing 1,900 adversarial prompts across harmful content, CBRN, and CSEM categories on seven state-of-the-art vision-language and audio-language models. Using simple perceptual transformations (FigStep-Pro, Intelligent Masking, and audio perturbations), the authors demonstrate attack success rates exceeding 75% for specialized domains, revealing that models with near-perfect text safety (0% ASR) become highly vulnerable to cross-modal attacks.


**Strengths**

- Demonstrates real, practical vulnerabilities of frontier multimodal systems.
- Covers multiple models, modalities, and high-risk categories with a unified evaluation pipeline.
- Low-skill transformations (typography, pitch shifts) achieve high ASR, underscoring practical risk.
- Attack recipes and evaluation criteria are well documented for reproducibility (within ethical limits).
- The accessibility of the attacks highlights immediate and practical security threats in widely deployed commercial systems.


**Weaknesses**

- Most methods are adaptations of existing techniques (e.g., FigStep-Pro extends FigStep) rather than fundamentally new attack mechanisms.
- Results lack confidence intervals, multiple runs, and human validation; reliance on a single GPT-4.1 judge may introduce bias.
- Provider-specific vulnerabilities are noted but not analyzed, leaving it unclear whether failures arise from OCR/ASR limits, safety filters, or encoder behavior.
- Proposed countermeasures (cross-modal checks, adversarial training) are not experimentally tested, offering little quantitative guidance for mitigation.
- Attack set favors transformations already known to succeed, is mostly English-only for audio, omits other modalities such as video, and provides no negative or real-world prevalence sampling.
- Identifies provider-specific vulnerabilities (Llama-4's visual weakness vs. Gemini's audio weakness) but doesn't explain underlying mechanisms.




**Suggestions**

- Add confidence intervals, multiple experimental runs, and statistical significance tests. Validate the GPT-4.1 judge with a human-annotated subset and include benign control inputs to rule out generic failure modes.
- Confirm that transformed images/audio convey the intended harmful meaning to human observers.
- Quantitatively test proposed defenses such as cross-modal consistency checks, perceptual anomaly detection, and adversarial training against the presented attacks.
- Extend audio tests to low-resource languages and accents, and explore additional modalities (e.g., video, 3D) or adaptive multi-modal attacks.
- Provide compute/resource details and release a staged or sanitized benchmark/code package to enable safe replication and follow-up research.



**Spelling, Formatting and Grammar Issues**

1. Line 22: "boarder semantic-level reasoning" should be "**broader** semantic-level reasoning"
2. Line 295: "The disparity between text safety (0" - appears to be cut off mid-sentence
3. Inconsistent hyphenation of "FigStep-Pro" vs "FigStep_pro" throughout the paper